# Strain-level characterization of broad host range mobile genetic elements transferring antibiotic resistance from the human microbiome

Samuel C. Forster [1,2,3,4✉], Junyan Liu[1,4], Nitin Kumar [1], Emily L. Gulliver [2,3], Jodee A. Gould[2,3], Alejandra Escobar-Zepeda[1], Tapoka Mkandawire [1], Lindsay J. Pike[1], Yan Shao[1], Mark D. Stares[1], Hilary P. Browne [1], B. Anne Neville[1] & Trevor D. Lawley [1✉]

Mobile genetic elements (MGEs) carrying antibiotic resistance genes (ARGs) disseminate ARGs when they mobilise into new bacterial hosts. The nature of such horizontal gene transfer (HGT) events between human gut commensals and pathogens remain poorly characterised. Here, we compare 1354 cultured commensal strains (540 species) to 45,403 pathogen strains (12 species) and find 64,188 MGE-mediated ARG transfer events between the two groups using established methods. Among the 5931 MGEs, we find 15 broad host range elements predicted to have crossed different bacterial phyla while also occurring in animal and environmental microbiomes. We experimentally demonstrate that predicted broad host range MGEs can mobilise from commensals *Dorea longicatena* and *Hungatella hathewayi* to pathogen *Klebsiella oxytoca*, crossing phyla simultaneously. Our work establishes the MGE-mediated ARG dissemination network between human gut commensals and pathogens and highlights broad host range MGEs as targets for future ARG dissemination management.

[1] Host-Microbiota Interactions Laboratory, Wellcome Sanger Institute, Hinxton CB10 1SA, UK. [2] Centre for Innate Immunity and Infectious Diseases, Hudson Institute of Medical Research, Clayton, Vic 3168, Australia. [3] Department of Molecular and Translational Science, Monash University, Clayton, Vic 3800, Australia. [4] These authors contributed equally: Samuel C. Forster, Junyan Liu. ✉email: sam.forster@hudson.org.au; tl2@sanger.ac.uk

Humans are colonized by microbial communities dominated by bacteria from the Firmicutes, Bacteroidetes, Actinobacteria, and Proteobacteria phyla that play an essential role regulating immune response[1], aiding sustenance[2], and providing pathogen colonization resistance[3]. Antibiotic treatment, though intended to eliminate pathogens, simultaneously eradicates indigenous commensal bacteria that are sensitive to the antibiotic. This can result in a microbiome with a vastly altered community structure and function; however, commensal species with intrinsic or acquired antibiotic resistance are protected from elimination. Antibiotic selection likely results in antibiotic resistance genes (ARGs) accumulation among commensal bacteria[4] that may also act as a reservoir from which ARGs are transferred on mobile genetic elements (MGEs) or by bacteriophage transduction to other species, including pathogens, via horizontal gene transfer (HGT)[5–12].

The extent of HGT in human gut microbiome and the types of MGEs involved, especially those associated with ARGs, have been the focus of continued research interest in the last decade. Several key metagenomic studies showed that HGT between gut bacteria is more frequent than HGT with bacteria from other body sites or environments because intestinal bacteria occupy the same habitat[5]. Most MGEs involved mobilise within the same phylum or lower taxonomic groups[9] and the transfer of ARGs between pathogens and commensals is considered limited[5,13]. A recent bioinformatic study utilising two separate genome collections has demonstrated a capacity to accurately predict compatible HGT host-recipient pairs[12]. However, experimental validation of HGT has largely relied on high-throughput chromatin conformation capture (Hi-C) which has uncovered extensive HGT in situ including between pathogenic and commensal species[10,11]. Many studies using animal models have also demonstrated individual cases of increased HGT between pathogens and commensals during infections[14,15]. Despite these advances, the majority of large-scale studies still do not experimentally validate MGE mobility at the isolate level. It is therefore imperative to understand the true scale of HGT between pathogenic and commensal species coexisting in the human gut, to identify MGEs posing high risks in order to better inform future interventions to curb spread of ARGs. We and others have recently demonstrated that the vast majority of the human gastrointestinal bacteria can be cultured[16–21].

In this work, we apply this resource to systematically investigate the extent of HGT between pathogens and commensals, with a focus on ARG-associated MGEs. We are able to confirm host range of MGEs with high confidence, strain-level resolution and validate in vitro putative past HGT events to demonstrate that these MGEs retain the ability to mobilise and can potentially spread associated ARG to numerous bacteria species.

## Results

**MGEs are shared extensively between commensals and pathogens.** To map the extent of horizontally shared MGEs that carry ARGs between pathogenic and commensal bacteria of the human gastrointestinal tract, we first compared 1354 commensal genomes (530 species) from the Human Gastrointestinal Microbiota Genome Collection (HGG)[22] (Supplementary Data 1) to 45,403 publicly available genomes (Supplementary Data 2) selected to capture representatives of 12 gastrointestinal pathogenic and opportunistic pathogenic species which will be referred to as pathogens in this work, including 8 Proteobacteria; *Klebsiella oxytoca* ($n = 139$), *K. pneumoniae* ($n = 7712$), *Escherichia coli* ($n = 17,142$), *Salmonella enterica* ($n = 10,394$), *Shigella sonnei* ($n = 1290$), *S. flexneri* ($n = 453$), *Campylobacter coli* ($n = 919$) and *C. jejuni* ($n = 1554$) and 4 Firmicutes; *Enterococcus faecalis* ($n = 1364$), *E. faecium* ($n = 1706$), *Clostridioides difficile* ($n = 2016$) and *Clostridium perfringens* ($n = 138$) (Fig. 1a). Some of the 12 species are among the most prevalent gastrointestinal pathogens globally[23–25], some are posing great risk to the public as they become increasingly resistant to antibiotics[26].

We reasoned that genes originating either directly or indirectly through recent horizontal transfer events would exhibit high nucleotide homology between isolates incongruent with phylogenetic distance. Pairwise gene comparisons were performed between the 1354 commensal genomes and 45,403 pathogen genomes to identify those genes sharing significant nucleotide identity (>99% identity across over 500 bp in organisms <97% 16S rRNA homology[5]) between strains from either group. This analysis found 389,541 putative horizontally transferred genes (Fig. 1b).

To further identify genes encoding antibiotic resistance within the 389,541 putatively transferred genes, we computationally defined ARGs by comparison to the Comprehensive Antibiotic Resistance Database (CARD)[27]. This identified 64,188 (16.5%) of the putative horizontally acquired genes shared between commensals and pathogens to be ARGs. These ARGs were dominated by multidrug efflux complexes, aminoglycoside resistance, cationic antimicrobial peptide resistance, and beta-lactamases (Supplementary Fig. 1). Notably, we observed no statistically significant enrichment in ARG class or gene families in either pathogen or commensal genomes. While these results reflect that pathogen-associated genes dominate ARG databases[27,28], they also suggest no obvious barriers to ARG dissemination within either pathogens or commensals and between the two groups. This highlights the capacity of gut commensals to act as a reservoir, where ARG could be transferred into as well as from transiently colonising pathogens. These observed patterns suggest that ARG dissemination networks are highly interconnected within the gut microbiome[5] and further demonstrate that they transcend the type of bacterial symbiosis with humans.

To define the MGEs responsible for mediating ARG transfer events between pathogens and commensals, we next consolidated the 64,188 shared ARGs into the common genetic elements by combining elements with greater than 90% shared homology across the element. A total of 5931 MGEs were identified within the dataset through this process. Analysis of these elements demonstrated a range of ARGs including dihydrofolate reductases, tetracycline resistance and aminoglycoside resistance with no enrichment for any individual class relative to the occurrence within the larger dataset.

The pathogens *E. faecalis* ($n = 1364$ genomes), *C. difficile* ($n = 2016$ genomes) *and E. faecium* ($n = 1706$ genomes) from the Firmicutes phylum shared the greatest number of distinct genetic elements with the commensal isolates with significant enrichment by Fisher exact test relative to numbers of genomes ($q < 10^{-6}$, $q < 10^{-4}$ and $q < 10^{-6}$). In contrast, the pathogens from the Proteobacteria phylum including, *Escherichia coli* ($q < 10^{-12}$; $n = 17,142$ genomes), *K. pneumoniae* ($q < 10^{-6}$; $n = 7712$ genomes) and *Shigella sonnei* ($q < 10^{-3}$; $n = 1290$ genomes), were significantly under-represented by Fisher Exact Test when evaluating the overall number of shared distinct MGEs with the commensal microbiota. Equally, commensal Bacteroidetes and Actinobacteria shared few ARG-containing MGEs with any of the pathogenic species considered in this study. While no events were observed with Fusobacteria, the small number of genomes limited statistical interpretation. Thus, although ARG-associated MGEs are widespread, MGE diversity and abundances are not evenly distributed across gut bacterial phyla, with Firmicutes exhibiting the highest enrichment of MGE diversity ($p < 0.05$; Fisher Exact Test) and sharing the most MGEs between commensals and pathogens ($p < 0.05$; Fisher Exact Test).

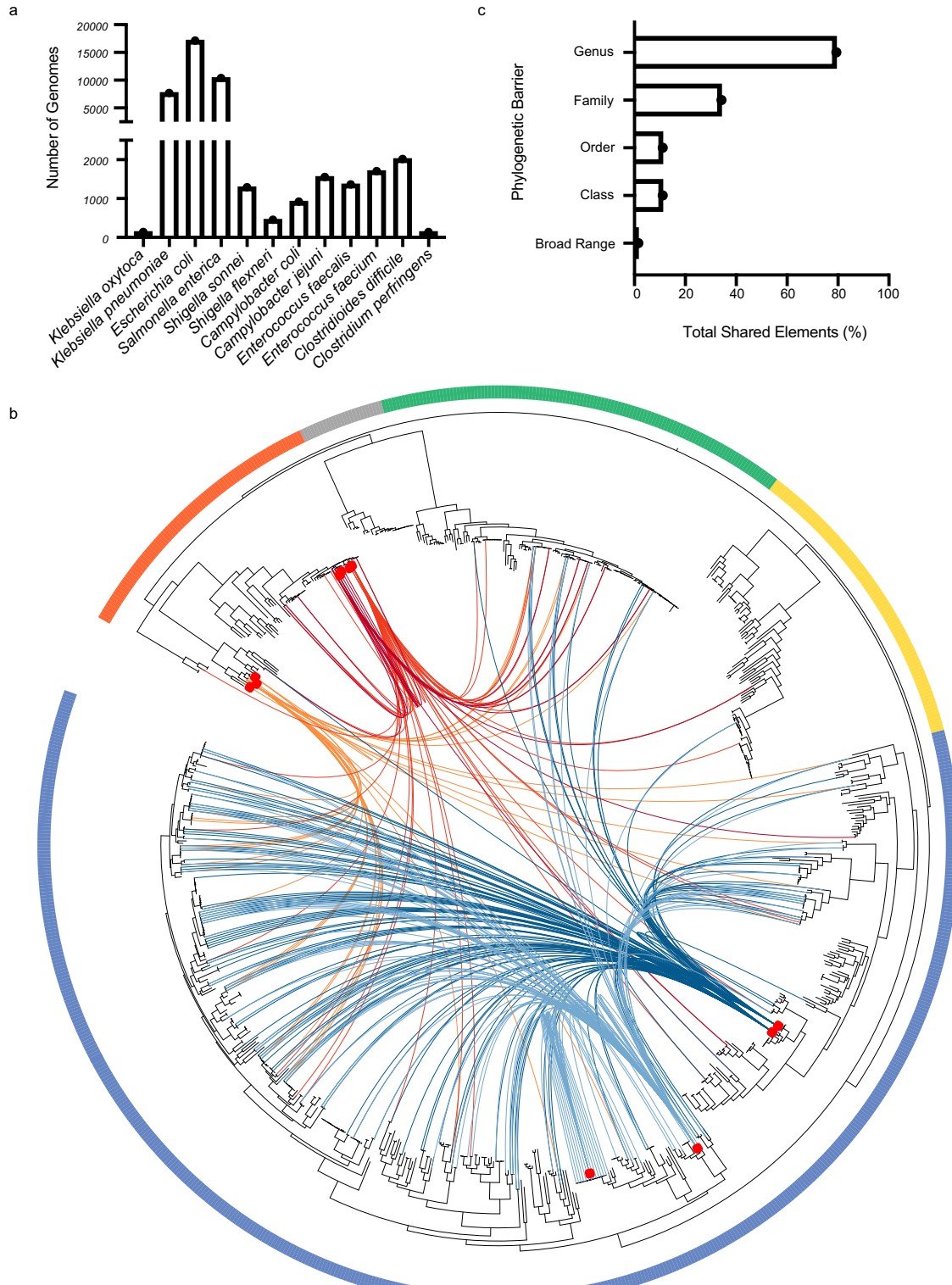

**Fig. 1 Patterns of horizontally transferred genetic elements between gastrointestinal commensal and pathogen species. a** Number of genomes from each of the twelve pathogenic species included in the analysis. **b** Actinobacteria (Gold), Bacteroidetes (Green), Firmicutes (Blue), Fusobacteria (Grey) and Proteobacteria (Orange) are indicated by the surrounding border. Red boxes indicate location of pathogens on the phylogenetic tree. Connecting lines indicate shared putative horizontally transferred genes. Links from *Escherichia* (Red), *Klebsiella* (Dark Red), *Campylobacter* (Orange), *Enterococcus* (Dark Blue), and *Clostridia* (Light Blue) pathogens are shown. **c** Phylogenetic barriers of MGE demonstrating inter-genus (79.3%), inter-family (34.2%), inter-order (11.14%), inter-class (11.12%) and broad host range, inter-phyla (1.5%) horizontal transfer.

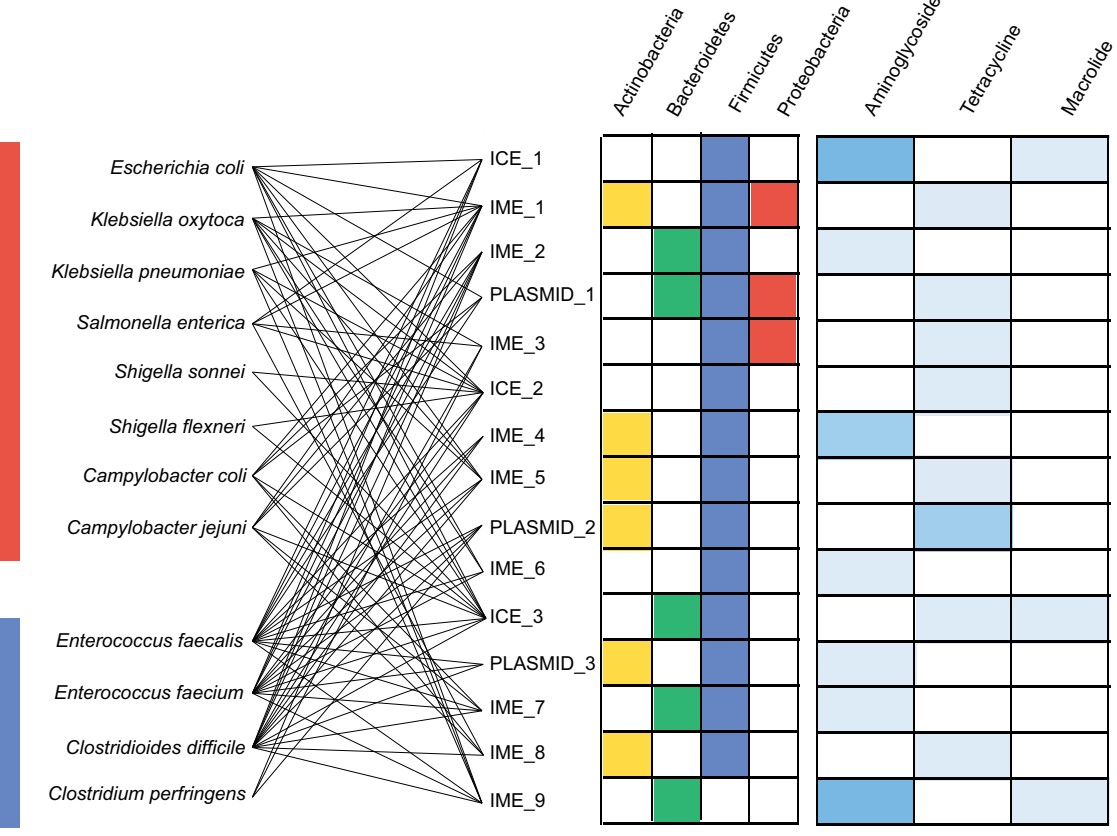

**Fig. 2 Taxonomic distribution and functional composition of 15 broad host range mobile genetic elements.** Occurrence of 15 promiscuous elements shared between pathogenic bacteria and commensal Actinobacteria (yellow), Bacteroidetes (green), Firmicutes (blue), and Proteobacteria (red). The presence of 1 (light blue), 2 (moderate blue) or 3 (dark blue) ARGs (Aminoglycoside, Tetracycline, Macrolide). Connecting lines show the presence of promiscuous elements in respective pathogenic bacteria.

**Specific MGEs are predicted to have transferred between phyla.** Our ability to confidently assign specific MGEs to bacterial host species provided a unique opportunity to generate a phylogenetic framework to study MGE host range. It is known that the horizontal transfer range of a MGE is limited by phylogenetic barriers of the bacterial hosts[8], although this has not been investigated at large scale across human gut bacteria. We found that 79.33% of MGEs transferred across genera, 11% transferred across classes and, interestingly, 1.5% of MGEs had a broad host range being found across multiple phyla (Fig. 1c). Our results suggest the widespread presence of phylogenetic barriers which prevent the majority of MGEs disseminating broadly across bacterial phyla, with the majority of ARG transfer being restricted to within a bacterial genus.

Given the obvious risks to human health associated with the capacity to disperse between bacterial phyla, we focused our analysis on those MGEs capable of broad host range and wide-scale dissemination between pathogens and commensals inferred from diverse phylogenetic occurrence. This analysis identified 15 broad host range MGEs shared between more than one bacterial phylum within our dataset. To provide a greater understanding of the genomic architecture surrounding the identified elements we coupled long-read sequencing technology with automated and manual annotations to generate high-quality reference genomes for the 15 broad host range MGEs (Supplementary Data 6).

The 15 promiscuous MGEs include three plasmids (PLAS-MID_1-3), three integrative and conjugative elements (ICEs; ICE_1-3), and nine integrative and mobilizable elements (IMEs; IME_1-9) (Supplementary Data 3). While the majority ($n = 14$) of these elements could be found in commensal Firmicutes, six

were also found in commensal Actinobacteria, five in commensal Bacteroidetes, and three in commensal Proteobacteria (Fig. 2). Elements occurred in a median of 5 of the 12 pathogenic species within this study (41.6%; min: 3; max: 10) with elements in three Firmicutes (75%; min = 1, max = 4) and two Proteobacteria (25%; min = 0, max = 7) species. Between one and four ARGs were found on each element, with tetracycline and aminoglycosides being the most common encoded resistance. There were no common genetic elements that provide a universal explanation for the observed broad host range.

We next classified the broad host range MGEs in the context of previously reported incompatibility (Inc) groups[29–31] based on the type of MGE replication machinery. We examined each element for the presence of a gene encoding RepA protein using PSI-BLAST (cutoff E-value<1e-05). Our analysis identified putative RepA proteins in eight MGEs that share limited similarity (between 20 and 40%) with known Inc groups, and one element, PLASMID_1 (Supplementary Fig. 2), that contains a putative RepA which is not closely related to any known Inc group (Supplementary Data 4). Thus, nine MGEs have putative novel replication proteins, but no identifiable genes encoding RepA could be recognized in the remaining six elements (Supplementary Data 3). We found that four elements (IME_1, IME_5, ICE_3, IME_8) encode RepA proteins that belong to a clade including the RepA found in IncQ plasmids and one element (PLASMID_2) that encodes a RepA that belongs to a clade that contains RepA found in IncA/C plasmids (Fig. 3a). In another cluster, RepA proteins from another three elements (ICE_1, ICE_2, IME_4) share ancestry with RepA from IncP1 plasmids (Fig. 3b). Importantly, MGEs belonging to IncA/C,

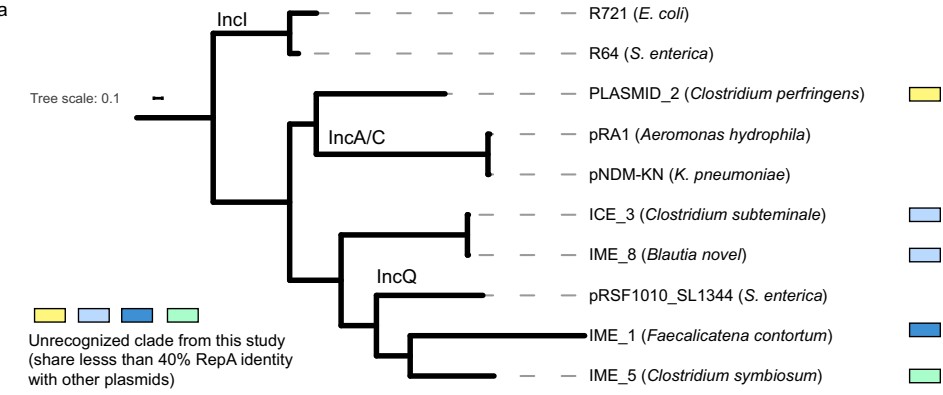

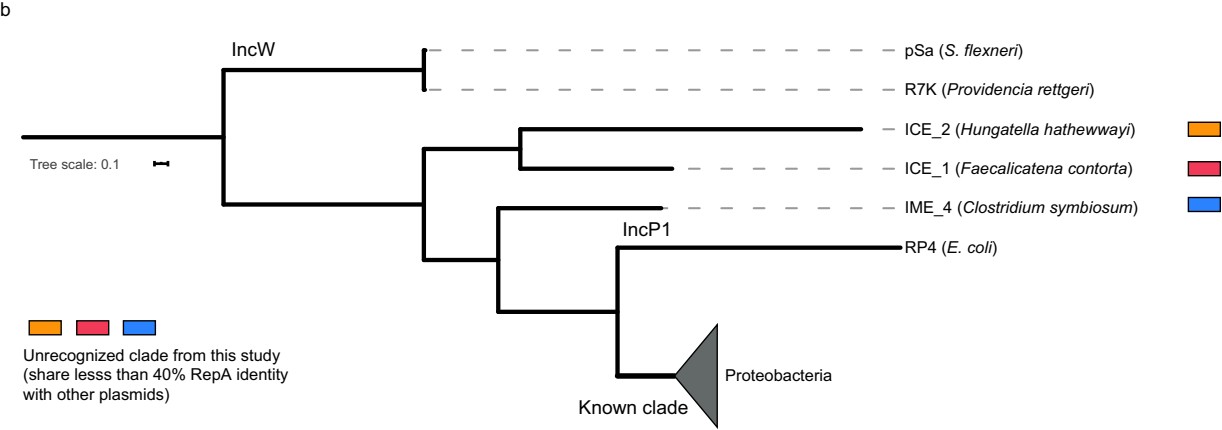

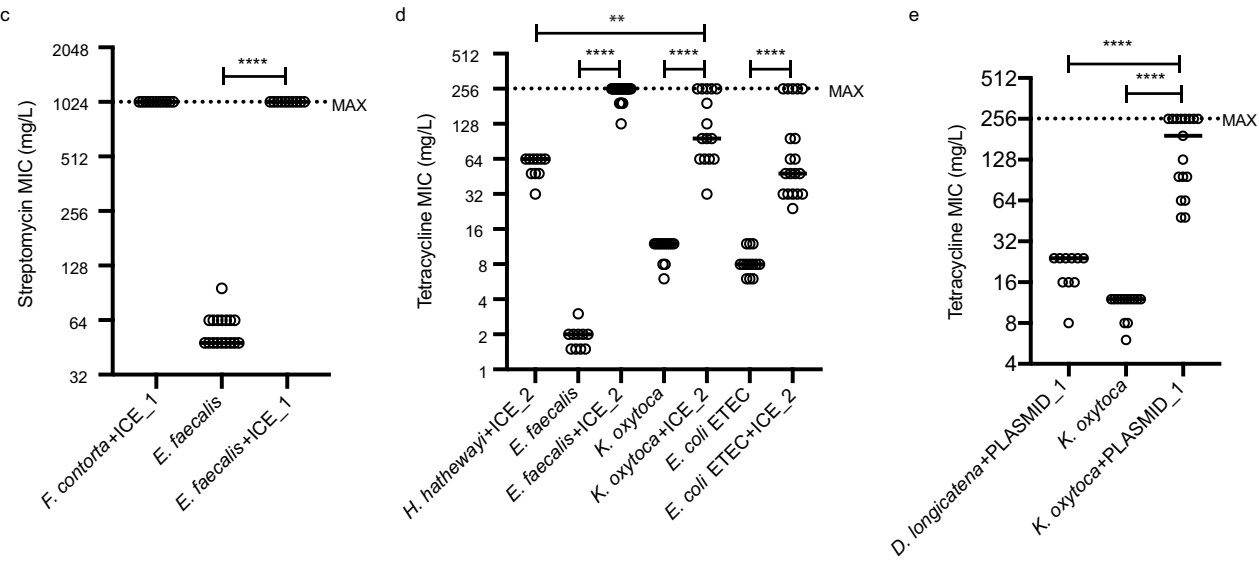

IncQ and IncP1 groups commonly carry ARGs in Proteobacteria[20,21] but MGEs with related replication machineries have not been previously recognized as drivers of broad host range ARG dissemination in Firmicutes and Bacteroidetes from the human microbiome.

**Broad host range MGEs are mobilizable in vitro.** Despite the observed wide host range of these ARG-associated MGEs based on our genomic analysis, it is essential to experimentally determine if these MGEs remain active. We next used in vitro conjugation assay to test the ability of these MGEs, especially ICEs

**Fig. 3 Classification of broad host range mobile genetic elements and experimental validation of horizontal transfer between distantly related species.**
**a**, **b** Phylogeny of the RepA proteins from 8 elements with representatives from closely related incompatibility groups. **c** ICE_1 from *F. contorta* can conjugate into *E. faecalis* ($n = 11, 17, 10$; two-tailed Mann–Whitney test, ****$p$-value = 0.000000118535588). **d** ICE_2 from *H. hathewayi* can conjugate into *E. faecalis*, *K. oxytoca* and *E. coli* ETEC ($n = 10, 10, 30, 13, 15, 13, 19$; two-tailed Mann–Whitney test, left to right, ****$p$-value = 0.000000001179718, 0.000000026707861, 0.000000005757490, **$p$-value = 0.001316401326497). **e** plasmid PLASMID_1 from *D. longicatena* can be transferred into *K. oxytoca* by conjugation ($n = 10, 13, 17$; two-tailed Mann–Whitney test, left to right, ****$p$-value = 0.000000237071175, 0.000000016700088). The median of each data set is indicated by a short horizontal line. The maximal antibiotic concentration of E-strips (MAX) is indicated by the dotted line. Source data are provided as a Source Data file.

and plasmids, to mobilise between commensals and pathogens of different classes or phyla. ICE_1 carries *aadK* (streptomycin resistance) and can be found in a *Faecalicatena contorta* strain (Clostridia class of Firmicutes). ICE_2 carries *tetM* (tetracycline resistance) and can be found in a *Hungatella hathewayi* strain (Clostridia class of Firmicutes). Both ICEs can be transferred into a streptomycin and tetracycline sensitive *E. faecalis* strain (Bacilli class of Firmicutes) by dry patch conjugation, demonstrating their ability to mobilise across different classes at different frequencies (Fig. 3c, d). Based on earlier analysis of this study, ICE_2 is predicted to be able to mobilise across phyla as well. We further selected a plasmid PLASMID_1 which carries a poorly characterised tetracycline resistant determinant and can be found in a *Dorea longicatena* strain (Lachnospiraceae family of Firmicutes). Using filter and dry patch conjugations respectively, ICE_2 and PLASMID_1 were transferred into a tetracycline sensitive *Klebsiella oxytoca* strain (Enterobacteriaceae family of Proteobacteria) (Fig. 3c, e). Additionally, ICE_2 can conjugate into another Enterobacteriaceae pathogen *E. coli* ETEC H10407 which is tetracycline sensitive (Fig. 3d). These results demonstrated that the ICE and plasmid still retain the ability to mobilise between different bacterial phyla. In all five pairs of conjugation assays, transfer of the MGE of interest into recipient bacterial strains was confirmed by PCR on all transconjugants (Supplementary Fig. 4; Supplementary Fig. 5) and long-read sequencing of some randomly selected transconjugants (Supplementary Data 5). Conjugation efficiency is expressed as frequency of antibiotic-resistant transconjugants as the rate of gaining resistance by natural mutation is significantly lower or even below detection limit (Supplementary Fig. 3). Interestingly, we noted that formerly tetracycline sensitive *K. oxytoca* strain, after receiving either PLASMID_1 or ICE_2, exhibit higher MIC (minimal inhibitory concentration) than the original donor strains (Fig. 3d, e). This implies the same ARG containing MGE can be responsible for quantitatively variable antibiotic resistant phenotypes in different bacterial host species within the gastrointestinal microbiota.

**Broad host range MGEs are prevalent outside the human gut**. To understand the prevalence and environmental range of the 15 broad host range elements we next examined 4349 high coverage human gastrointestinal microbiome-associated metagenomes available within the HPMC database[32]. We determined prevalence rates of these elements to be between 0.52% and 98.2% with no statistically significant enrichment observed with Fisher Exact Test based on element type, size or antibiotic resistance genes carried (Fig. 4). To understand if these elements or associated homologues are also found within the microbiome communities of other human body sites, we determined the occurrence and prevalence of these elements within the skin, nasal cavity and female reproductive tract datasets of the Human Microbiome Project dataset. This analysis identified five elements limited in distribution to the human gastrointestinal tract, eight were detected in samples from the human nasal cavity, nine were detected in samples from the human vagina and three were detected in samples from the human skin with a coverage of 99%

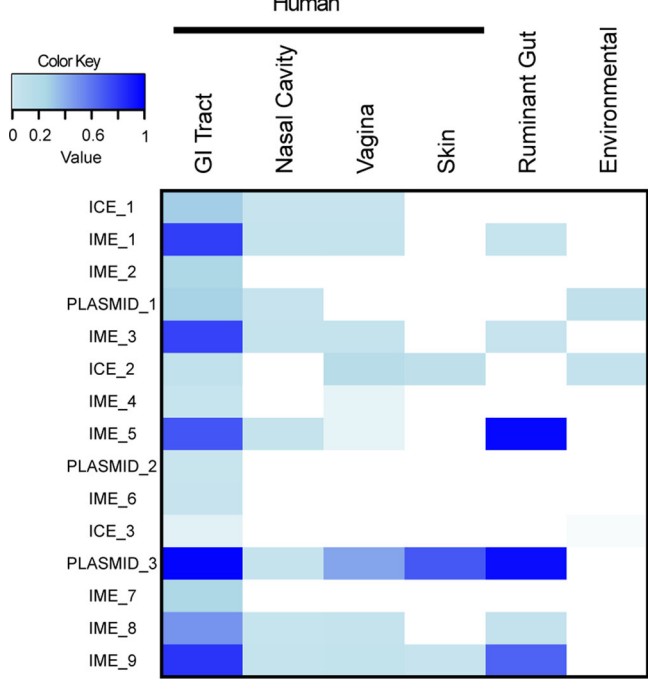

**Fig. 4 Prevalence of the promiscuous mobile elements in human body, ruminant gut, and environmental microbiomes.** Prevalence of promiscuous elements across the human gastrointestinal tract, nasal cavity, vagina, skin, ruminant gut, and environmental samples. Blue represents the frequency with which an element was detected within a metagenomic sample derived from the site. No detection is represented in white.

and detection rate of 0.001% (Fig. 4). Considered together, these results suggest an interconnection within the human microbiome, as the occurrence of certain MGE across different body sites is a result of its host's movement potentially disseminating ARG in the process. Equivalent analysis of samples from ruminant gastrointestinal microbiome identified the presence of six elements, all of which can be also detected in both human gut, nasal cavity, and vagina. On the other hand, only three elements were detected within environmental soil samples (Fig. 4). Hence, we demonstrate the presence of identical broad range MGEs in humans, animals, and environmental sources, highlighting the need for a One Health approach to understand ARG distribution.

## Discussion
While the prevalence of ARGs within human gut microbiome has been well established, we are lacking an understanding of ARG-associated MGE host range and prevalence in the context of the entire human microbiome. Our work represents large-scale, whole-genome, strain-level analysis utilising unsupervised discovery coupled with experimental validation to generate an MGE-mediated ARG dissemination network between human commensals and pathogens. Despite the variable quality of publicly available genomes and limitation imposed by database for ARG

identification, this work characterizes the diversity and host range of MGEs harboured by the gut microbiome, demonstrates the retained ability of key broad host range MGEs to mobilize between diverse commensal and pathogenic species. In a medical context, these broad host range MGEs may represent a significant threat aiding ARG dispersal independently of the infection control measures established to contain specific pathogens. These insights suggest effective antimicrobial stewardship will require a focus not only on controlling antibiotic-resistant pathogens but tracking, managing, and limiting ARG-associated MGE dissemination[33] from both pathogenic and commensal bacterial species.

## Methods

**Bacterial culturing**. Culturing was performed under anaerobic conditions (BOC; 290564-L) in a Whitley A95 anaerobic workstation (Don Whitley) using YCFA media at 37 °C. Selective culturing was performed anaerobically and aerobically using YCFA media supplemented with antibiotic at indicated concentration at 37 °C.

**Bacterial conjugation and antibiotic sensitivity testing**. Donor and recipient isolates were grown in YCFA broth anaerobically overnight without shaking. For filter conjugation, donor and recipient cultures were diluted to $OD_{600} = 1$, then mixed in 4:1 ratio and immobilized onto an autoclaved 0.45 μm mixed cellulose ester membrane (Whatman; 10401606) which was then placed on a YCFA plate with the side containing bacterial cells facing up. For dry patch conjugation, donor and recipient cultures were concentrated to $OD_{600} = 40$ and $OD_{600} = 10$ respectively, mixed by equal volume (4:1 ratio by cell density), and spotted on YCFA plate. All conjugation plates were incubated anaerobically overnight and washed with 1-1.5 mL PBS to harvest cells. The cell suspension was plated on YCFA plates with or without antibiotic for transconjugants selection and enumeration respectively. The concentration of streptomycin (AppliChem; A1852) used was 300 μg/mL. The concentrations of tetracycline (Apollo Scientific; BIT0150) used was 50 μg/mL (low selection) and 60 μg/mL (high selection). These plates were incubated aerobically to select against obligatory anaerobic donors. All the recipient controls were treated following the same protocol respectively. The MIC of specific antibiotic against donor, recipient, and transconjugants were measured using E-strips (bioMerieux; 526800; 522518) according to manufacturer's instructions on YCFA media. Donor isolates for ICE_1 (DSM 108236; JCM 31258), PLASMID_1 (DSM 108233) and ICE_2 (CCUG 68736) are available in public depository.

**Polymerase chain reaction (PCR) for mobile element detection**. PCR was performed to confirm transfer of mobile element into recipient bacteria. ICE_1 forward primer (TGATATCATGGAAGGTCGGCA) and reverse primer (ACCGCCCTGAACAATTGATG) target a 181 bp region within the *aadK* gene. PLASMID_1 forward primer (AAAGCAGCTATCATTCCGGGT) and reverse primer (TGCCCGCCTTTGAAGATACC) target a 301 bp region within the *mobC* gene. ICE_2 forward primer (TTGATGCCCTTTTGGAAATC) and reverse primer (ACTGCATTCCACTTCCCAAC) target a 294 bp region within *tetM* gene.

**Statistical tests**. Two-tailed Mann–Whitney test (unpaired, nonparametric) was used to compare MIC or conjugation frequency data where applicable.

**Short read genome sequencing and annotation**. Genomic DNA was extracted from bacterial cell pellets using a phenol-chloroform method. In a MaXtract high density phase lock gel (PLG) tube (Qiagen; 129056), mix 3 ml of RNAse (Thermo fisher Scientific; 12091-021) treated cell lysis with 5 ml of phenol: chloroform: isoamyl alcohol (25:24:1) and centrifuge for 5 min at 2800 rpm in an Eppendorf centrifuge 5810 R (Eppndorf) to separate the phases. Transfer the top aqueous phase to a fresh PLG tube and repeat the previous step. Mix the aqueous phase with 5 ml of chloroform: isoamyl alcohol (24:1) in a fresh PLG tube and repeat the centrifugation. Repeat this step. Mix the final aqueous phase with 2.5 volumes of cold 100% ethanol to precipitate the DNA. Libraries were prepared and paired end-sequencing performed on the Illumina Hi-Seq platform with a target library fragment size of 450 bp and a read length of 150 bp at the Wellcome Sanger Institute according to standard protocols. Annotated assemblies were produced using the pipeline standard Wellcome Sanger Institute prokaryotic assembly and annotation pipeline[34]. Briefly, for each sample an optimal assembly was created using Velvet v1.2[35] and VelvetOptimiser v2.2.5, scaffolded using SSPACE v3.0[36], and sequence gaps filled using GapFiller v1.10[37]. PROKKA v1.14[38] was used for automated gene annotation. Genome sequences have been deposited in the European Nucleotide Archive.

**Long read genome sequencing and annotation**. Genomic DNA was extracted from bacterial cell pellets using MasterPure™ complete DNA purification kit

(Lucigen; MC85200). Oxford Nanopore Minion Sequencing, multiplexed libraries were sequenced using the ligation sequencing kit according to manufacturer's instructions. Where required, genomes were demultiplexed using guppy v1.1 and assembled using canu v1.9[39]. Within genomes mobile genetic elements were located by the ARG previously identified. The element boundaries were determined as a consensus from predictions by IslandViewer 4[40], ICEfinder v1.0[41], Alien_-Hunter v1.7[42], and the presence of inverted and direct repeat sequences.

**Phylogenetics and reference genomes**. The phylogenetic analysis was conducted by extracting amino acid sequence of 40 universal single copy marker genes[43,44] from bacterial collection using SpecI v1.0[45]. The protein sequences were concatenated and aligned with MAFFT v. 7.20[46], and maximum-likelihood trees were constructed using raxML v8.2.11 with default settings. All phylogenetic trees were visualized in iTOL v5.6[47]. Average nucleotide analysis (ANI) was calculated by performing pairwise comparison of genome assemblies using MUMmer v3.0[48]. Genomes from 12 pathogenic species were obtained from NCBI based on taxonomic classification.

**Horizontally acquired gene analysis**. Shared genes were identified by pairwise BLAST v2.6.0+ of annotated genes (greater than 99% identity across 500 bp or greater sequence). Putative horizontally acquired genes were identified by shared gene in organisms with greater than 97% 16S rRNA homology[7]. The CARD database version 1.9 was used to identify ARG with a 95% identity and 90% coverage cutoff.

**Mobile element identification**. Complete mobile element sequences were retrieved from the PLSDB plasmid database v. 2019_06_03[49] and the ICEberg 2.0 database. Mash v2.1.1 sketches were generated (-i -S 42, -k 21 –s 1000) and putative mobile elements screened with Mash screen (-v 0.1, –i 0.95). Elements with less than 950 of 1000 shared Mash hashes was considered as different elements.

**Replication initiation protein (RepA) classification**. RepA proteins were identified by PSI-BLAST v2.6.0 + (Position-Specific Iterated BLAST) against a plasmid replication protein database created from three studies[50–52] with a cutoff E-value of 1e-05 and RepA protein sequences available on NCBI.

**Metagenomic prevalence**. Prevalence of mobile elements was assessed using bowtie2 v2.3.4.1[53] searches of metagenomic datasets retrieved from the HPMC database[32] and the HMP database[22]. Elements were considered present within the sample when occurring at greater than 0.001% of sample with greater than 99% coverage.

**Reporting summary**. Further information on research design is available in the Nature Research Reporting Summary linked to this article.

## Data availability

The sequence data generated in this study have been deposited in the ENA database under accession codes PRJEB37690 and PRJEB49256. The MIC, ARG counts, and conjugation frequency data generated in this study are provided in the Source Data file. The sequence data from the HGG used in this study are available in the ENA database under accession codes ERP105624 and ERP012217. Source data are provided with this paper.

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

## Acknowledgements

This work was supported by the Wellcome Trust [206194]; the Australian National Health and Medical Research Council [1091097, 1159239, and 1156333 to SCF], and the Victorian Government's Operational Infrastructure Support Program (SCF). The authors would also like to acknowledge the support of the Wellcome Sanger Institute Pathogen Informatics Team and Monash University eResearch. Funding for open access charge: Wellcome Sanger Institute.

## Author contributions

S.C.F., B.A.N., J.L., N.K., and T.D.L. conceived the study. S.C.F., E.G., N.K., J.A.G., T.M., and A.E. performed the computational analysis. S.C.F., B.A.N., L.J.P., Y.S., H.P.B., J.A.G., and M.D.S. isolated and purified the bacteria and performed genome sequencing. J.L., T.M. performed the experimental analysis. All authors read and contributed to the manuscript. S.C.F. and J.L. contributed equally.

## Competing interests

T.D.L. is the co-founder of Microbiotica Limited. The authors declare no other competing interests.
