## [Peer Review File · Nature Communications]

REVIEWER COMMENTS

Reviewer #1 (Remarks to the Author):

In their paper, “In vitro demonstration of broad host range mobile genetic elements transferring antibiotic resistance from the human microbiome”, Forster and colleagues analyzed a set of available genomes for highly similar antibiotic resistance genes and similar mobile elements with attention to their taxonomic distributions. They then analyzed the mobile genetic elements containing these antibiotic resistance genes for similarities in mobilization machinery and found 15 different ‘broad host range’ elements. They then conjugated three isolates with different mobile elements and showed that they retained the ability to transfer into *E. faecalis* and *K. oxytoca*.

While I appreciate the conjugation experiments that the authors performed, the experiments in the manuscript are extremely limited in scope. The authors test the conjugation of 3 strains (each with their own ICE or plasmid) with two additional bacterial strains. I was expecting a much broader set of experiments with deeper analysis.

The analysis of broad host range elements is very cursory. Contrary to the authors claim that they are the first paper to do such an analysis, there are tons of other papers that have explored broad host range plasmids and ICEs—their genetics and their phylogenetic range—and other papers which have explored the distribution of mobile elements in microbiomes.

Given the limited scope of work in the paper, the authors make some large claims. For example, I do not think that the authors showed that these transfers are not ancient (lines 174-175), that their results highlight phylogenetic barriers (135-137), and an interconnectedness of the human microbiome (lines 210-213). Each of these claims requires careful analyses, many of which have been performed previously.

For the analysis that was there, the computational methods were not very detailed.

Reviewer #2 (Remarks to the Author):

Lawley and team sought to investigate the scale of HGT between pathogens and commensals in our human gut. The team compared the antimicrobial resistance genes (ARGs) and MGEs in HGT from databases containing gut commensals genomes (1354 genomes of 530 cultured species) with 45,403 genomes of 12 enteropathogens from public databases in order to estimate the magnitude of these HGT events and to identify the major MGEs players. In the global problem of antimicrobial resistance, this is an important fundamental question to raise and to elucidate. These enteropathogens are important carriers of ARGs among human pathogens and the frequency and/or prevailing HGT mechanisms/MGEs of these and their respective presence in the gut commensals will contribute to our understanding of their inter-relationships between pathogens and commensals. In addition, the host range of MGEs at strain-level and validation of these events to demonstrate the MGEs' ability to mobilise and spread ARGs across numerous bacterial species were also elucidated. The study also demonstrated the presence of phylogenetic barriers which prevent MGEs from disseminating broadly across bacterial phyla. Fifteen prevalent MGEs were identified and its prevalence and environmental range of these broad host elements were examined from high coverage gastrointestinal microbiome- metagenomes datasets (HPMC database). These 15 promiscuous MGEs highlighted will be important focus and pave the way for future studies to understanding their contribution to HGT events other animal hosts and the environment. This is a very interesting approach to tackle the AMR and how the aggregated large datasets might be exploited to understand this problem.

Some limitations of the study:

- (1) The directionality of transfer of ARGs between commensals and pathogens cannot be ascertained.
- (2) Bacteriophages play an important role in the HGT of genes in our gut microbiota. While the study is confined to bacterial commensals, prophages and bacteriophages' contribution may not be taken into consideration in the overall MGE events.
- (3) The dataset is based on public databases of genome data on 12 specific enteropathogens. The magnitude of HGT and the MGEs highlighted would be based on the quality of the reads in the database (which may have a wide range of variability using different sequencing platforms and conditions, with bias in the sequencing especially involving short reads, repeated sequences and incomplete sequences esp with plasmids) as opposed to using more curated complete genomes. As such, it may reflect an underestimation of the HGT events and this may be explained and discussed further in the manuscript.
- (4) The ARGs and MGEs in gut commensals may not be well characterized and established (and limited by use of any ARG databases). This is a limitation not from the study perspective, but from our knowledge gaps, which given time will improve and with data which validate the ARGs in our gut commensals.

Reviewer #3 (Remarks to the Author):

I find the presence of potential broad range plasmids and mobile genetic elements interesting, but I found the experimental setup and concept slightly confusing. As I understand it, the original concept was to identify exchange of mobile genetic elements and antibiotic resistance between commensal and pathogenic gut microbiota. The concept of pathogens and commensal, however, are ill-defined. For instance, *Escherichia coli* Nissle 1917 is probiotic bacterium, but the authors define *E. coli* as pathogens. Although genetic exchange across divergent phylogroups are highly interesting, I really do not see the importance in the context of the gut microbiota, with much more frequent transfer across closely related strains. I also find it very strange to use *Klebsiella oxyto* as a recipient strain, as this is not a common pathogen in the human gut, as least to my knowledge. Having said that, I find the results intriguing. However, I would have liked to see some more documentation connected to the conjugation experiments. For instance, what is the conjugation frequency. I think it also would have been useful with presenting complete genome sequencing results of some of the transconjugants.

Detailed comments

l. 55-57. I think the concept of pathogens and commensals are ill defined

l. 63-64. I would not consider the current study as large-scale with respect to lab experiments.

l. 65-67. I cannot see that the study addresses this issue

l. 87-89 The study should have addressed experimentally the frequency of transfer.

Suppl Fig. 4 If a 1 kb ladder was used I cannot see the correspondence with the proposed PCR product size, as described in l. 297 to 307. Actually, I think it would have been better to show results from sequencing than a PCR product sizes.

Reviewer comments:

We would like to thank all the reviewers for taking the time to review the manuscript. We appreciate their insightful comments and suggestions that have certainly assisted in improving the overall quality of the manuscript.

Reviewer #1 (Remarks to the Author):

In their paper, “In vitro demonstration of broad host range mobile genetic elements transferring antibiotic resistance from the human microbiome”, Forster and colleagues analyzed a set of available genomes for highly similar antibiotic resistance genes and similar mobile elements with attention to their taxonomic distributions. They then analyzed the mobile genetic elements containing these antibiotic resistance genes for similarities in mobilization machinery and found 15 different ‘broad host range’ elements. They then conjugated three isolates with different mobile elements and showed that they retained the ability to transfer into *E. faecalis* and *K. oxytoca*.

While I appreciate the conjugation experiments that the authors performed, the experiments in the manuscript are extremely limited in scope. The authors test the conjugation of 3 strains (each with their own ICE or plasmid) with two additional bacterial strains. I was expecting a much broader set of experiments with deeper analysis.

It should be noted the establishment and optimization of each pairwise experimental system for conjugation are highly technical, requiring substantial and intensive effort with these commensal bacteria. As such this work represents the first validation of in vitro conjugation for all 3 donor species presented and includes combination of donor, MGE, recipient that are completely novel. For each assay it is necessary to screen 100’s of potential recipients *E. faecalis* and *K. oxytoca* strains for appropriate antibiotic sensitivities then to optimize antibiotic concentrations to eliminate the likelihood of recipient strains gaining resistance through spontaneous mutation. Despite this, in response to your comment we have now included additional conjugation experiments demonstrating further MGE transfer with additional species (Figure 3d)

The analysis of broad host range elements is very cursory. Contrary to the authors claim that they are the first paper to do such an analysis, there are tons of other papers that have explored broad host range plasmids and ICEs—their genetics and their phylogenetic range—and other papers which have explored the distribution of mobile elements in microbiomes.

We apologize for the ambiguity of this statement. Our intention was to convey the point this was the first study to combine experimental validation with computational analysis. This work, coupled with the genome sequenced culture collection, provides the capacity to not only perform unsupervised discovery of MGE aided ARG transfer between commensal and pathogen species in the human gut microbiome but coupled this with targeted experimental validation to demonstrate these relationships represent active, biologically relevant relationships. We have rephrased to clarify the message (lines 27, 229-232)

Given the limited scope of work in the paper, the authors make some large claims. For example, I do not think that the authors showed that these transfers are not ancient (lines 174-175), that their results highlight phylogenetic barriers (135-137), and an interconnectedness of the human

microbiome (lines 210-213). Each of these claims requires careful analyses, many of which have been performed previously.

(i) Our intention with this statement was to demonstrate that the elements responsible for transfer of the observed ARGs remain active within the specific commensal bacterial host rather than being a remnant element that has lost the ability to transfer. We have now revised the relevant sentence to clearly highlight this point (lines 176-178)

(ii) Our analysis clearly demonstrates a predominance of shared elements consistent with phylogenetic structure; however, we have tempered the language and replaced “demonstrate” with “suggest” given the challenge of directly proving this relationship without large scale experimental validation that is beyond the scope of this research (line 138).

(iii) As we and many others have previously shown there is a strong interconnectedness and movement of isolates between human body sites. The purpose of this statement was to highlight the capacity for these events to also share ARGs. We have now modified this sentence to improve clarity (line 217-220).

For the analysis that was there, the computational methods were not very detailed.

We apologise for the brevity, we have now added additional details to the relevant computational methods section (lines 347-372)

Reviewer #2 (Remarks to the Author):

Lawley and team sought to investigate the scale of HGT between pathogens and commensals in our human gut. The team compared the antimicrobial resistance genes (ARGs) and MGEs in HGT from databases containing gut commensals genomes (1354 genomes of 530 cultured species) with 45,403 genomes of 12 enteropathogens from public databases in order to estimate the magnitude of these HGT events and to identify the major MGEs players. In the global problem of antimicrobial resistance, this is an important fundamental question to raise and to elucidate. These enteropathogens are important carriers of ARGs among human pathogens and the frequency and/or prevailing HGT mechanisms/MGEs of these and their respective presence in the gut commensals will contribute to our understanding of their inter-relationships between pathogens and commensals. In addition, the host range of MGEs at strain-level and validation of these events to demonstrate the MGEs' ability to mobilise and spread ARGs across numerous bacterial species were also elucidated. The study also demonstrated the presence of phylogenetic barriers which prevent MGEs from disseminating broadly across bacterial phyla. Fifteen prevalent MGEs were identified and its prevalence and environmental range of these broad host elements were examined from high coverage gastrointestinal microbiome- metagenomes datasets (HPMC database). These 15 promiscuous MGEs highlighted will be important focus and pave the way for future studies to understanding their contribution to HGT events other animal hosts and the environment. This is a very interesting approach to tackle the AMR and how the aggregated large datasets might be exploited to understand this problem.

Some limitations of the study:

(1) The directionality of transfer of ARGs between commensals and pathogens cannot be ascertained.

A key point of this study is the demonstration that these MGEs are capable of readily transferring ARGs both from the pathogens to commensals and from commensals to pathogens. In this context the ancient directionality of the original event is of less significance than the capacity of the element to continue to actively transfer. We have now included a sentence to clearly highlight this relationship and also mention the role commensal could play as a ARG reservoir. (line 82-83). Notably, this bi-directional nature of transfer is experimentally demonstrated in those elements that are validated.

(2) Bacteriophages play an important role in the HGT of genes in our gut microbiota. While the study is confined to bacterial commensals, prophages and bacteriophages' contribution may not be taken into consideration in the overall MGE events.

This is certainly an important point. While this study was focused on the role of MGEs we have now included a sentence in the introduction to recognize the numerous mechanisms through which this might occur including bacteriophages (line 50).

(3) The dataset is based on public databases of genome data on 12 specific enteropathogens. The magnitude of HGT and the MGEs highlighted would be based on the quality of the reads in the database (which may have a wide range of variability using different sequencing platforms and conditions, with bias in the sequencing especially involving short reads, repeated sequences and incomplete sequences esp with plasmids) as opposed to using more curated complete genomes. As such, it may reflect an underestimation of the HGT events and this may be explained and discussed further in the manuscript.

Thank you for this suggestion, the quality of the genome sequences used from pathogens is obviously a potential limitation that could constrain the ability to detect all the HGT events. We have now added a statement to make this point clear to the reader in the discussion (line 232-233). This stated, it should be noted that despite this limitation we have still identified and experimentally validated key MGEs with the capability described.

(4) The ARGs and MGEs in gut commensals may not be well characterized and established (and limited by use of any ARG databases). This is a limitation not from the study perspective, but from our knowledge gaps, which given time will improve and with data which validate the ARGs in our gut commensals.

We agree this is an important point that will be addressed in future research and have added this to the conclusion (line 232-233)

Reviewer #3 (Remarks to the Author):

I find the presence of potential broad range plasmids and mobile genetic elements interesting, but I found the experimental setup and concept slightly confusing. As I understand it, the original concept was to identify exchange of mobile genetic elements and antibiotic resistance between commensal and pathogenic gut microbiota. The concept of pathogens and commensal, however, are ill-defined. For instance, *Escherichia coli* Nissle 1917 is probiotic bacterium, but the authors define *E. coli* as pathogens. Although genetic exchange across divergent phylogroups are highly interesting, I really do not see the importance in the context of the gut microbiota, with much more frequent transfer across closely related strains. I also find it very strange to use *Klebsiella oxyto* as a recipient strain, as this is not a common pathogen in the human gut, at least to my knowledge. Having said that, I find the results intriguing. However, I would have liked to see some more documentation connected to the conjugation experiments. For instance, what is the conjugation frequency. I think it also would have been useful with presenting complete genome sequencing results of some of the transconjugants.

While we agree all transfer events are potentially biologically interesting this work was specifically focused on identifying those elements capable of transferring functions, particularly antibiotic resistance, between phylogenetically diverse isolates. To establish a system for selection of transconjugants it was necessary to select recipients that exhibited both antibiotic sensitivity and aerobic growth capability. Likewise, it was necessary that the donor commensal strain was strictly anaerobic to enable negative selection with an aerobic environment. Given these limitations, selection of appropriate recipient strains was strongly constrained. We have now also included ETEC as a key pathogenic strain of *E. coli* to complement the existing data from *E. faecalis* and *Klebsiella oxytoca*, which itself is an emerging pathogen. We have also included associated conjugation frequency data as Supplementary Figure 3. All the sequence data has been deposited in ENA under project accession number PRJEB49256 for both the donor and recipient strains and these data are now included in Supplementary Table 5.

Detailed comments

I. 55-57. I think the concept of pathogens and commensals are ill defined

Given the context specific capacity for any isolate to act pathogenically the problem of definition of both pathogen and commensal remains problematic within the field more generally. We have now clarified the list within the manuscript (line 78-80) and have also included a clear list of the genomes in Supplementary Table 2.

I. 63-64. I would not consider the current study as large-scale with respect to lab experiments.

We apologize for the confusion; the key point here was to highlight that the bioinformatic analysis was large scale and this led to targeted experimental validation not include in many equivalent studies of this type.

I. 65-67. I cannot see that the study addresses this issue

While it is true this study does not completely address this issue and there remains substantial further research, we believe this study provides key understanding of specific MGEs capable of readily transferring ARGs between many diverse species. This study, while alone certainly not

sufficient to address this significant global health problem, nevertheless provides an important foundation to advance this cause.

I. 87-89 The study should have addressed experimentally the frequency of transfer.

Supplementary Figure 3 details the frequency of transfer events for each conjugation experiment. We have now included specific reference to this within the main text to clearly illustrate this point.

Suppl Fig. 4 If a 1 kb ladder was used I cannot see the correspondence with the proposed PCR product size, as described in I. 297 to 307. Actually, I think it would have been better to show results from sequencing than a PCR product sizes.

We apologise for the failure to label the DNA ladder. This has now been added to Supplementary Figure 4. We have also included product sizes from the methods to ensure all relevant information is available in the figure legend. We have also made all the sequencing data available through PRJEB49256 to facilitate access to this data as described above

REVIEWERS' COMMENTS

Reviewer #2 (Remarks to the Author):

The authors have addressed the questions raised by the reviewers and the revised version of the manuscript is an improvement overall. There is added laboratory data to the conjugation experiments, including the use of a pathogenic *E. coli* and details of the transconjugant frequencies. Some rephrasing in the text also improved the clarity of the discussion.

The findings are significant and demonstrated predicted broad host range MGEs can mobilise ARGs from commensals to pathogenic organisms and across phyla.

Reviewer #3 (Remarks to the Author):

I think you have adequately addressed my previous concerns.